# Severe COVID-19 Pneumonia in a Three-Year-Old with Congenital Iron and B12 Deficiency Anemia of Unknown Etiology: A Case Report

**DOI:** 10.3390/children10040616

**Published:** 2023-03-24

**Authors:** Theodore Daniel Liapman, Jurijs Bormotovs, Dace Reihmane

**Affiliations:** 1Department of Human Physiology and Biochemistry, Riga Stradins University, LV-1007 Riga, Latvia; 2Altnagelvin Area Hospital, Western Health and Social Care Trust, Derry BT47 6LS, UK; 3Children’s Clinical University Hospital, LV-1004 Riga, Latvia

**Keywords:** COVID-19, pneumonia, anemia, biomarkers, pediatrics

## Abstract

Since COVID-19 first emerged in Wuhan, China, and was declared a global pandemic by the WHO, researchers have been meticulously studying the disease and its complications. Studies of severe COVID-19 disease among pediatric populations are scarce, leading to difficulty in establishing a comprehensive management approach. Case presentation: This report outlines a case of a long-standing combined iron and vitamin B12 deficiency anemia in a three-year-old treated at the Children’s Clinical University Hospital due to severe COVID-19 disease. The patient’s clinical condition coincided with the derangement of biomarkers described in the literature, including lymphopenia, increased neutrophil/lymphocyte ratio (NLR), decreased lymphocyte/C-reactive protein ratio (LCR), as well as elevated inflammatory markers such as CRP and D-dimers. The patient developed severe bilateral pneumonia requiring invasive ventilation, high-flow oxygen, immunosuppressive therapy with dexamethasone and tocilizumab, and supplementation of anemia deficits with blood transfusion and vitamin B12 administration. Conclusions: Our findings are consistent with the most important biomarkers reported in the literature indicative of severe disease progression. Additionally, poorly controlled anemia may be suggested as a potentially important risk factor for severe COVID-19 disease among children. However, additional quantitative research is required to establish the nature and severity of the risk.

## 1. Introduction

In December 2019, the first of a series of respiratory tract infection cases of what was then, of unknown origin from Wuhan, Hubei province, China, was reported to the WHO [1]. Due to the rapid spread of the virus (R0 = 2.5 [2]), on 11 March 2020, COVID-19 was declared a global pandemic by the World Health Organization (WHO) [3]. The culprit for the disease was identified to be a novel zoonotic coronavirus and was named SARS-CoV-2, similar to SARS-CoV-1, which was responsible for the 2002–2004 outbreak, and to MERS-CoV, outbreaks of which were identified mainly in Saudi Arabia and the Republic of Korea [4]. As of February 2023, the ongoing pandemic has infected over 670 million people worldwide, with close to 7 million deaths [5].

Studies have shown that in most patients (81%), disease presentation is mild to moderate and non-life-threatening, with a mortality rate of 2.3% [6]. In moderate disease presentation, affecting ~40% of those infected, patients may show clinical signs of pneumonia (fever, dyspnea, and fast breathing). Patients presenting with a more severe course of disease additionally demonstrate a respiratory rate of over 30 breaths/minute, severe respiratory distress, or an SpO_2_ < 90% on room air [7].

These severe presentations typically fall into acute respiratory distress syndrome (ARDS) and refractory hypoxemia, evident by the low saturation rate. Patients may experience respiratory failure, organ damage, and dysfunction leading to sepsis and a corresponding increase in fatality rate [6].

In children, infection more often results in mild disease compared with adult populations [6] despite being the most tested age group with the highest number of positive cases [8]. Symptoms of COVID-19 infection among pediatric populations appear to develop in significantly lower proportions along with reduced severity (only 7% of cases are characterized as severe compared to a much higher 26% rate in adults) [9].

There are multiple explanations in the literature as to why the course of the disease appears milder in children than in adults. The existing explanations attribute the lower incidence of severe disease presentation among children to the lower prevalence of risk factors such as pre-existing cardiovascular disease, chronic kidney disease, chronic lung diseases, diabetes mellitus, hypertension, immunosuppression, obesity (BMI > 30), sickle cell disease, and pregnancy [10] as well as to the fact that innate immune responses decline with age [9]. However, there may be another explanation, one presented by Bunyavanich et al. following a study in which they examined nasal epithelium for asthma biomarkers [11]. The study suggests that adults have higher expression compared to children of angiotensin-converting enzyme 2 (ACE2), which plays a role in the mechanism by which the virus enters the infected cell [1].

Nevertheless, some children do present with severe disease. According to a meta-analysis published by Graff et al. in 2021, the risk factors for severe COVID-19 in children include a history of obesity, asthma, obstructive sleep apnea, diabetes or pre-diabetes, and baseline oxygen requirement. The study also links increased risk for severe COVID-19 to pulmonary, gastrointestinal, endocrine, neurologic, and psychiatric disease, immunocompromising conditions, and history of pre-term birth. Similarly to Cui et al., the same study reports limited availability of pediatric data regarding severe COVID-19 disease [12].

Anemia is linked with a severe course of the disease, with most studies conducted in adults and only a small percentage in pediatric populations [13]. We present a case of a three-year-old child who developed severe COVID-19 pneumonia requiring admission to the pediatric intensive care unit at the Children’s Clinical University Hospital (CCUH) in Riga, Latvia. During the course of the disease, the child developed left-sided pneumothorax and required invasive ventilation as well as aggressive IV drug therapy via a centrally placed line. The child’s previous medical history is notable for an iron and B12 anemia of unknown etiology that was poorly managed before admission but is otherwise unremarkable.

This clinical case report aims to add to the body of evidence demonstrating the link between iron deficiency and B12 anemia and a more severe course of COVID-19 infection.

## 2. Methods

This study was designed as a retrospective case report. The case was analyzed using data collected from the patient’s electronic and physical medical records and demographic characteristics. Doctors’ notes, patient charts, imaging, laboratory investigations, and treatment records were studied and compared with the available literature. As a retrospective case analysis, its limitations are dictated by the available case data.

Consent for the implementation of the study was obtained via a consent form signed by the patient’s mother. The study design was approved by Riga Stradins University’s ethics committee and the Children’s Clinical University Hospital. The report does not contain any personally identifiable information as per the General Data Protection Regulation (GDPR) guidelines.

## 3. Case Presentation

A 3-year-old girl was admitted to the Children’s Clinical University Hospital (CCUH) in November of 2021. The patient’s medical history was notable for a previous hospitalization due to right-sided pneumonia at the age of 19 months, poorly controlled combined iron and B12 deficiency anemia, and a light form of atopic dermatitis. The girl had been attending kindergarten and had been vaccinated according to the national vaccination calendar. At the time, a vaccine against SARS-CoV-2 was not yet available for children. The patient’s father suffered from multiple allergies and allergic rhinitis, and the patient’s mother had a mild form of psoriasis.

On admission, the patient complained of a three-day history of epigastric abdominal pain, dry cough, fatigue, and a fever reaching 39 °C. On examination, the skin was normal with physiological colour and moist mucous membranes. There was slight hyperemia of the throat but no lymphadenopathy. The abdomen was soft, non-tender and active peristalsis was heard on auscultation. Although the patient was reluctant to drink water, her appetite appeared normal. The patient’s urine output was normal.

A rapid antigen test performed on admission was SARS-CoV-2 positive. This was confirmed with PCR testing and led to a diagnosis of COVID-19. Over the following four days, the patient’s condition worsened with increasing fatigue, cough, abdominal pain, and tachypnea (reaching 100/min), dropping SpO_2_ (87–90% on room air). A chest X-ray obtained on day 2 (Figure 1) revealed multiple infiltrates present bilaterally, more pronounced on the left lung than the right, consistent with bilateral pneumonia and a capillary blood gas (CBG) test revealed type I respiratory failure (RF).

Due to the worsening symptoms, the patient was placed on non-invasive ventilation, and immunosuppressive therapy was provided with 10 mg of dexamethasone once daily (OD). Due to concerns for a superinfection empiric antibiotic therapy was provided as well with 500 mg amoxicillin three times daily (TDS). To address the patient’s low hemoglobin level (6.8 g/dL) caused by the long-standing anemia, a blood transfusion was provided, and hemoglobin levels increased to 10 g/dL before settling at 9 g/dL.

The patient’s clinical condition continued to decline with increasing pediatric warning score (PEWS) levels (Figure 2). Initially, attempts were made to mitigate the low SpO_2_% levels by increasing the oxygen flow rate from 2 L/min to 8 L/min and switching from a nasal cannula to a non-rebreather mask (NRM) as well as switching from amoxicillin to 800 mg cefotaxime TDS and further increasing immunosuppression by adding 200 mg tocilizumab IV (single dose). Nevertheless, the patient declined further, developing type 2 RF on CBG and hyperglycemia (21.3 mmol/L). The patient was therefore transferred to the pediatric intensive care unit (PICU) and placed on invasive ventilation.

Following intubation, a left-sided apical pneumothorax was discovered, most likely an iatrogenic complication resulting from the intubation. It was drained, and no further complications were observed.

Five days following admission to the CCUH, the patient was deemed septic due to the presence of a high fever (>38.5 °C), tachycardia (~150 bpm), tachypnea (prior to intubation), low leukocyte count (3.33 × 10^3^), and the presence of an active SARS-CoV-2 infection. Antibiotic coverage was increased with 800 mg cefotaxime TDS, 300 mg meropenem TDS, and 90 mg fluconazole OD.

Following the measures taken, the patient’s condition improved with increasing SpO_2_% levels, resolving respiratory failure on CBG, and gradually clearing signs of pneumonia on the chest X-ray. To prevent respiratory alkalosis, ventilation was switched to CPAP at this point and later on, the patient was extubated.

Following extubation, the patient’s breathing rate was normal at ~16/min. Initially, the patient required oxygen support to maintain SpO_2_% levels but was gradually weaned off until it was no longer required. A chest X-ray obtained at this stage (Figure 3) demonstrated resolving pneumonia. The patient was then discharged from the PICU and recovered on the ward with normalization of PEWS before discharge (Figure 2). The key interventions provided during the hospitalization period are shown in Table 1.

The patient was reviewed at the hospital one month following discharge, at which point complete resolution of respiratory symptoms was noted along with normal appetite, vital signs, and physical examination.

## 4. COVID-19 Biomarkers

Although no overt leukocytosis was observed, a relative increase in leukocyte levels (Figure 4) was seen. Lymphopenia was observed (Figure 4) in the first few days following admission but resolved two days before clinical improvement was noted. A marked increase in the neutrophil/lymphocyte ratio (Figure 5) was observed on day 4, along with C-reactive protein levels (Figure 6). The lymphocyte/CRP ratio (Figure 6) remained markedly low until day seven. At that point, it gradually increased concurrently with clinical improvement.

Other biomarkers that were associated with COVID-19 severity were measured as well. D-dimer and lactate dehydrogenase levels both remained elevated throughout the hospitalization period. Interleukin-6 (IL-6) was measured once on day five following admission, before admission to PICU, and was significantly elevated at 80.5 pg/mL.

## 5. Discussion

Since the SARS-CoV-2 virus first appeared in Wuhan, China, researchers around the globe have been meticulously studying the disease, its culprit, treatment approaches, complications, and long-term effects. Recent research has demonstrated a link between anemia and severe COVID-19 disease presentation. However, these studies were primarily conducted with adult cohorts or even excluded pediatric populations entirely [14,15,16]. However, although it has been shown that children tend to experience a milder course of the disease when compared with adult populations [6], this case highlights that severe COVID-19 may also occur in pediatric populations.

The most important biomarkers reported in the literature as indicative of severe disease progression were present in this case. Lymphopenia is associated with a nearly threefold increase in the risk of poor outcomes [17]. Elevated NLR, CRP, and low LCR have all been linked to complications of COVID-19 [18]. Elevated D-dimers and lactate dehydrogenase are both associated with a threefold and fivefold increase in the risk for poor outcomes respectively [17]. Elevated interleukin-6 (IL-6) has also been linked with severe complications [19]. This may be a result of hyperactivation of the humoral immune pathway triggered by severe COVID-19 disease, marking the dysregulation in the host’s immune response to the infection [20].

These abnormal changes progressed as the patient’s clinical condition declined, and their return to baseline levels coincided with clinical improvement. Our findings support using these biomarkers as adjuncts to clinical assessment in predicting the course of the disease.

The patient in this case report was admitted with combined iron deficiency and B12 anemia. It is known that iron and B12 deficiencies result in low levels of hemoglobin and red blood cells, leading to a decrease in their oxygen-carrying capacity [21]. In patients with severe COVID-19 pneumonia, such as the one presented in this case study, this decreased oxygen-carrying capacity is detrimental to patient survival, especially in light of the hyper-metabolic state caused by the infection [14,15]. Moreover, it has been suggested in the literature that SARS-CoV-2 can lower functional hemoglobin levels and interfere with iron transport and red blood cell production, further decreasing oxygen-carrying capacity and exacerbating anemia [22]. Therefore, anemia correction should be considered a priority in the treatment approach.

The presence of a vitamin B12 deficiency along with iron deficiency introduces a diagnostic challenge. Vitamin B12 deficiency is uncommon among pediatric populations, usually caused by a similar deficit in the mother during pregnancy, and is typically considered in young children presenting with megaloblastic anemia [23]. However, the deficiency in vitamin B12 was masked in this case, as in cases with similarly combined anemias found in the literature [24], by the presence of iron deficiency, leading to microcytosis and hypochromia. Such combined anemias are uncommon, even more so among pediatric populations [24].

Vitamin B12 is known to partake in metabolic activity, nucleotide synthesis, and DNA maintenance, making it essential not only for red blood cell production but also for the production of leukocytes, including lymphocytes [25]. As stated, lymphopenia has also been shown to result from severe COVID-19 disease due to the exhaustion of CD8+ T lymphocytes [26]. Patients with a lower baseline of vitamin B12 may therefore be at increased risk for a more severe course of the disease.

As the patient deteriorated and developed respiratory distress, dropping saturation levels on room air requiring oxygen supplementation, it became necessary to intubate the patient and provide external respiratory support with mechanical ventilation. This also highlights the usefulness of the pediatric early warning score (PEWS) in identifying deteriorating patients for prompt transfer to the pediatric ICU.

Empiric antibiotic coverage was provided in order to mitigate the risk of superinfection as studies have demonstrated an increased risk of superinfection among hospitalized COVID-19 patients [27]. Two immunosuppressive medications were used over the course of the disease. dexamethasone treatment is supported by studies demonstrating its effectiveness in moderate and severe COVID-19 cases in lowering dependence on ventilatory support [28]. Despite this, the patient’s condition deteriorated, and treatment was also attempted with tocilizumab, which showed promise in some studies [29,30,31]. Whether or not the clinical improvement observed in the days that followed resulted from the use of tocilizumab, or the blood transfusion, or would have occurred with no interference as the disease ran its course cannot be established in a single case report. Further studies on the use of tocilizumab in severe COVID-19 might reveal its role in treatment.

## 6. Conclusions

This case study highlights anemia as a significant risk factor for severe COVID-19 disease among children. Although further studies are required, clinicians should consider anemia testing in risk stratification, be vigilant in correcting iron and B12 deficiencies in children presenting with COVID-19, and consider other measures, such as blood transfusions, where appropriate.

The use of inflammatory markers in managing COVID-19 infections has been well documented. They should be used as adjuncts to clinical assessment to guide clinicians in dealing with severe cases in pediatric populations both in general and in cases such as those with concurrent iron and B12 anemia.

## Figures and Tables

**Figure 1 children-10-00616-f001:**
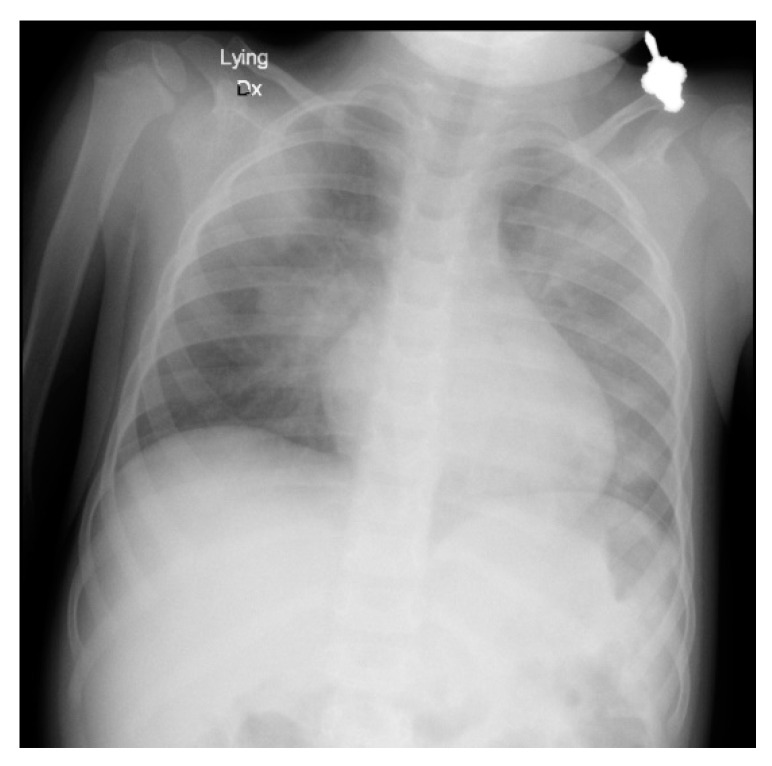
Day 2 Chest X-ray demonstrating bilateral pneumonia.

**Figure 2 children-10-00616-f002:**
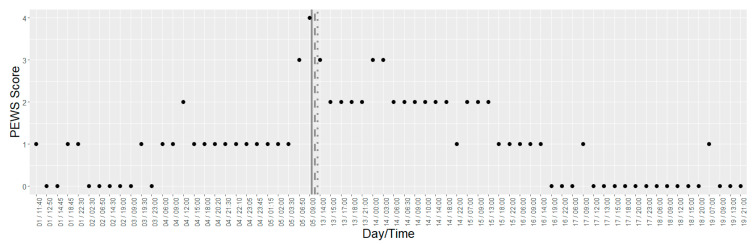
Pediatric Early Warning Score (PEWS) before and after PICU. The solid line denotes transfer to PICU + intubation, the dashed line denotes extubation, and the dotted line denotes discharge from PICU.

**Figure 3 children-10-00616-f003:**
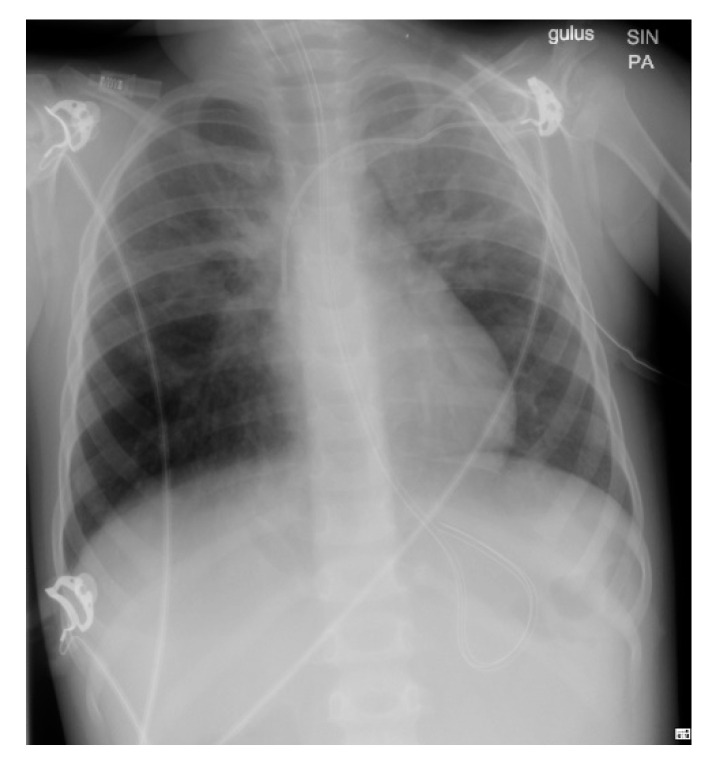
Day 2 Chest X-ray demonstrating decreased opacification bilaterally indicative of resolving pneumonia.

**Figure 4 children-10-00616-f004:**
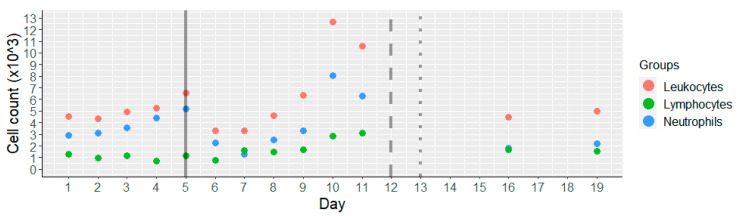
Leukocyte, lymphocyte, and neutrophil Counts (×10^3^) throughout the hospitalization period. The solid line denotes transfer to PICU + intubation, the dashed line denotes extubation, and the dotted line denotes discharge from PICU.

**Figure 5 children-10-00616-f005:**
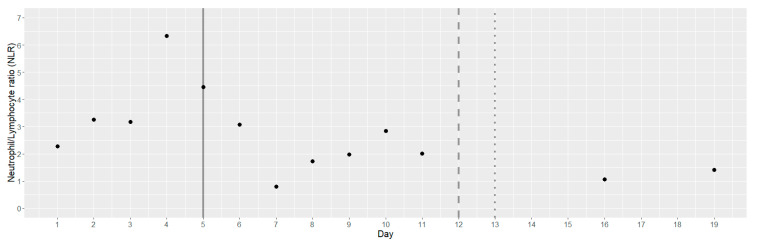
Neutrophil/Lymphocyte Ratio (NLR) throughout the hospitalization period. The solid line denotes transfer to PICU + intubation, the dashed line denotes extubation, and the dotted line denotes discharge from PICU.

**Figure 6 children-10-00616-f006:**
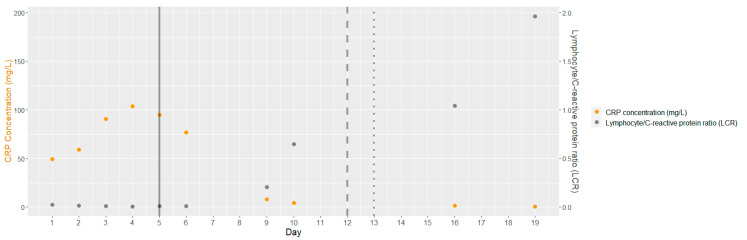
CRP (mg/L) and lymphocyte/CRP ratio levels throughout hospitalization. The solid line denotes transfer to PICU + intubation, the dashed line denotes extubation, and the dotted line denotes discharge from PICU.

**Table 1 children-10-00616-t001:** Manipulations and key medications administered throughout the hospitalization period.

Day	Manipulation	Medication
1—Onset		
3—Admission	Non-invasive ventilation Blood transfusion	Amoxicillin (day 3) → Cefotaxime (day 5) Dexamethasone
6—Transfer to PICU	Intubation + invasive ventilation	Cefotaxime Dexamethasone
7—Sepsis		Cefotaxime + Meropenem + Fluconazole
13—Improvement	Extubation → Non-invasive ventilation	
14—Transfer to ward	Non-invasive ventilation (Stopped on day 17)	
21—Discharge		

## Data Availability

The datasets generated and/or analyzed are not publicly available due patient confidentiality regulations but are available from the corresponding author on reasonable request.

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
