# Peer review of "Severe COVID-19 Pneumonia in a Three-Year-Old with Congenital Iron and B12 Deficiency Anemia of Unknown Etiology: A Case Report"

_children, 2023, doi:10.3390/children10040616_

Round 1
Reviewer 1 Report
Thank you for the opportunity to review this manuscript. It presents an important topic in the form of a case study on Severe Covid-19 pneumonia in a three-year-old with congenital 2 iron and B12 deficiency anemia of unknown etiology. Despite my wide appreciation to the authors, I recommend the following changes to the text.
1) Line 44 and 53: the period should be after the square brackets.
2) At the end of the introduction, please provide a unified purpose of the study, in the current form it is not there at all, and even the case reports have one.
3) A brief description of the case (socio-demographic data) should be included in the methodological study. In addition, this item also lacks a description of the procedures that were carried out in the case, such as imaging studies, COVID-19 testing, the fact of vaccination, etc. I would very much appreciate their completion.
4) I suggest including an abbreviated description of the procedures conducted day by day in tabular form.
5) The discussion is based on only 10 sources, with the rest of the paper citing only 27 items, which should necessarily be expanded to include new reports related to COVID. Particularly those talking about recommendations and recommendations for the treatment of severe covid disease.
6) The paper does not have a slant on its strengths and limitations (please supplement - because there are a lot of them!!!).
7) What are the conclusions of the study?
Greetings
Author Response
We thank you all for your valuable suggestions that have helped to improve our manuscript. Some of the suggestions from reviewers were contradictory (e.g. removing section of Methods vs adding more details in this section). We have reviewed some similar case reports published in Children and we hope that the manuscript is now structured as per the format for a scientific journal and that all of the parts are integrated in a way that the readers can understand. Once more, thank you for your contribution in the preparation of this manuscript. If you have any further comments / suggestions for improvement of our manuscript, we will be happy to address them.
Thank you for both the minor and the major suggestions. We've answered your comments below.
1) Line 44 and 53: the period should be after the square brackets.
Thank you for the remark. This suggestion has been implemented in manuscript.
2) At the end of the introduction, please provide a unified purpose of the study, in the current form it is not there at all, and even the case reports have one.
A unified purpose of the study has been formulated and added to the end of the introduction section, lines 84-86.
3) A brief description of the case (socio-demographic data) should be included in the methodological study. In addition, this item also lacks a description of the procedures that were carried out in the case, such as imaging studies, COVID-19 testing, the fact of vaccination, etc. I would very much appreciate their completion.
Thank you for the valuable suggestion. Unfortunately, little socio-demographic data is available from the hospital case file. The child attends kindergarten and had been up-to-date with her vaccinations, but had not received a COVID-19 vaccine as vaccination of paediatric population did not yet take place in Latvia at the time. This information has been added to the text (lines 103-105).
Regarding COVID-19 testing, this has been addressed in lines 113-114, a clarification was added that PCR testing was performed following the rapid antigen test to confirm the diagnosis.
The patient's last chest x-ray prior to discharge from the hospital, demonstrating resolving pneumonia, has been added and can be compared with the initial chest x-ray. The key interventions provided during the hospitalization period have been summarized in table 1 (line 170).
4) I suggest including an abbreviated description of the procedures conducted day by day in tabular form.
Thank you for the suggestion. A table summarizing the key interventions performed over the hospitalization period has been added (line 170).
5) The discussion is based on only 10 sources, with the rest of the paper citing only 27 items, which should necessarily be expanded to include new reports related to COVID. Particularly those talking about recommendations and recommendations for the treatment of severe covid disease.
One issue addressed in this case report which was one of the main motivating factors for the publication of this case report is the scarce available data regarding severe COVID-19 infections in children. This is both because severe COVID infections are much rarer in pediatric populations, a fact that is now elucidated more clearly in the discussion section in lines 205-208, and because the disease is still relatively new. Additionally, several points have been moved from the results section to the discussion section as they compare our findings with the literature and therefore belong in the discussion section. This brings up the number of references in the discussion section to 17 and the total number of references to 30 which is in line with number of references seen in other clinical case reports published in Children.
6) The paper does not have a slant on its strengths and limitations (please supplement - because there are a lot of them!!!).
Indeed, there are limitations to this study, the main one stems from its nature as a retrospective case report. The available information in the patient's case file is all that was available and no additional testing or examination can be done retroactively. A sentence to address this has now been added to the methods section (lines 91-92).
7) What are the conclusions of the study?
A conclusion section has now been added underneath the discussion to better communicate the conclusions of the study.

Reviewer 2 Report
I don't have suggestions. I considered that the paper is well documented and presented.
Author Response
The authors are grateful for the reviewer's time in reviewing this paper and for the positive feedback.

Reviewer 3 Report
I appreciate the effort the authors put into this case report. The case report provides some interesting observations regarding the clinical presentation and management of a pediatric patient with severe COVID-19. The authors presented a case of a three-year-old with long-standing combined iron and vitamin B12 deficiency anemia and severe COVID-19 disease, concluding that certain biomarkers are indicative of severe disease progression and that anemia is a significant risk factor for severe COVID-19 disease in this child. However, these biomarkers are known in severe COVID-19 disease, and the evidence for a correlation between anemia and severe COVID-19 disease in this case is weak.
The patient was previously admitted for right-side pneumonia at the age of 19 months. What was the underlying immunologic condition of the patient? Why was the combined iron and B12 deficiency anemia poorly controlled in this girl? If anemia is a risk factor for COVID-19 progression, why was the progression not prevented by the blood transfusion? Can you explain this condition?
Thank you.
Author Response
We thank you all for your valuable suggestions that have helped to improve our manuscript. Some of the suggestions from reviewers were contradictory (e.g. removing section of Methods vs adding more details in this section). We have reviewed some similar case reports published in Children and we hope that the manuscript is now structured as per the format for a scientific journal and that all of the parts are integrated in a way that the readers can understand. Once more, thank you for your contribution in the preparation of this manuscript. If you have any further comments / suggestions for improvement of our manuscript, we will be happy to address them.
The authors are grateful for the reviewer's report, which helps to improve the manuscript. We've answered your comments below.
1) The authors presented a case of a three-year-old with long-standing combined iron and vitamin B12 deficiency anemia and severe COVID-19 disease, concluding that certain biomarkers are indicative of severe disease progression and that anemia is a significant risk factor for severe COVID-19 disease in this child. However, these biomarkers are known in severe COVID-19 disease, and the evidence for a correlation between anemia and severe COVID-19 disease in this case is weak.
Thank you for this comment. The biomarkers included in the case report are indeed known in severe COVID-19. However, the available data on their behaviour in severe COVID-19 infections has mainly been collected from adult populations. Though in this case report no new biomarkers are highlighted, it does add to the small available data regarding existing biomarker behaviour in children with severe COVID-19. Additionally, we have changed the wording in the abstract to avoid misleading causal relations.
2) The patient was previously admitted for right-side pneumonia at the age of 19 months. What was the underlying immunologic condition of the patient?
The patient's immunologic background is only notable for a light form of atopic dermatitis (line 100). As for the right-sided pneumonia, we speculate that her underlying congenital combined B12 and iron deficiency anemia might have played a role in the need for her hospitalization then as well as the same proposed mechanism of reduced oxygen-carrying capacity could have influenced her symptoms then as well.
3) Why was the combined iron and B12 deficiency anemia poorly controlled in this girl?
According to the available information in the patient's file, the diagnosis of iron-deficiency anemia had been given long before the hospitalization addressed in this case report, though the B12 deficiency component of it was missing. We speculate that this might have been due to masking of the B12 deficiency features (high MCV and MCH) by the iron deficiency (low MCV and MCH). Nevertheless, we can't know the reason for certain as this information is not available to us.
4) If anemia is a risk factor for COVID-19 progression, why was the progression not prevented by the blood transfusion? Can you explain this condition?
We agree that a blood transfusion is indeed the best approach for correction of severe anemia as the one presented in this case. A blood transfusion was provided which brought the hemoglobin count in the patient up from 6.8g/dL to 9g/dL. This has been described in lines 128-131.

Reviewer 4 Report
The manuscript is generally well-written and the case is interesting.
Nevertheless, I would have the following suggestions in order to improve your manuscript:
- the section with material and methods should be removed
- in the results section you should not mention references, not even for justifying the treatment, you should move them to the discussions section
- please avoid non-scientific language like ''the patient was seen again''
- the discussions section should be expanded
- the conclusions should be mentioned in the different section
Author Response
We thank you all for your valuable suggestions that have helped to improve our manuscript. Some of the suggestions from reviewers were contradictory (e.g. removing section of Methods vs adding more details in this section). We have reviewed some similar case reports published in Children and we hope that the manuscript is now structured as per the format for a scientific journal and that all of the parts are integrated in a way that the readers can understand. Once more, thank you for your contribution in the preparation of this manuscript. If you have any further comments / suggestions for improvement of our manuscript, we will be happy to address them.
Thank you for your suggestions and feedback. Please see the answers we have provided below.
1) The section with material and methods should be removed
The methods section includes important information for the case report such as the source of patient data, the declaration regarding obtained consent, as well as the ethics committee approval. We would appreciate input from the journal editors regarding the need for this section.
2) In the results section you should not mention references, not even for justifying the treatment, you should move them to the discussions section.
Thank you for this comment. We wholeheartedly agree and have moved all discussion elements along with their references to the discussion section as per your recommendation.
3) Please avoid non-scientific language like ''the patient was seen again''.
Thank you for your comment. This sentence has been amended and now utilizes more appropriate language.
4) The discussions section should be expanded.
The discussion section has been expanded with a few elements elaborated upon. Please do let us know if you feel anything else is needed.
5) The conclusions should be mentioned in the different section.
Thank you, a conclusion section has been added underneath the discussion section (lines 267-275).

Round 2
Reviewer 4 Report
Thank you for improving your manuscript.
Nevertheless, I consider that the disccussions section should be further improved, and you should explain the relationship between anemia and COVID-19 complications based on the studies published so far.
I strongly recommend you to use a professional editing service for language issues since the language is not appropriate.
Author Response
Dear reviewer,
As per your suggestions, the discussion section has been expanded to further explain the relationship between anemia and COVID-19. Additionally, the manuscript has been subjected to English editing and should now include more appropriate language.
We appreciate your ongoing support.
Round 3
Reviewer 4 Report
No further comments.
Author Response
Thanks for reviewing